# Low-volume enrichment method supports high throughput bacteriophage screening and isolation from wastewater

Patrick O. Kenney[1], Oscar G. Gómez-Duarte[1,2]*

1 Division of Pediatric Infectious Disease, Department of Pediatrics, Jacobs School of Medicine and Biomedical Sciences, State University of New York (SUNY) at Buffalo, Buffalo, NY, United States of America, 2 Levine Children's Hospital, Atrium Health/Wake Forest University, Charlotte, NC, United States of America

* oscargom@buffalo.edu

**Data Availability Statement:** All relevant data are within the manuscript and its Supporting Information files. The URL containing supporting information is located at DOI: 10.17605/OSF.IO/AHGJF for open access. We have created a data set

## Abstract

Bacteriophage therapy is a rapidly growing field of study. Narrow host ranges, bacterial resistance, and limited antibiotic availability make lytic phages a feasible therapeutic potential. Phage discovery, a critical step in developing phage therapy, is a pathway to accessible treatment. This has always been a laborious, time-consuming and resource-intensive process. In this paper, we describe a 96-well plate low-volume bacteriophage enrichment method with concentrated environmental sources to rapidly discover and isolate phages targeting multiple organisms simultaneously. Samples from natural water sources, wastewater influent, and activated sludge were tested in large volume enrichment cultures and low-volume 96-well plate format. Each plate has the capacity to run as many as 48 different combinations with multiple bacterial hosts. The time to identify the presence of phage in a sample was 5 to 10 hours in the low-volume format versus a minimum of 2 days in the traditional enrichment method. The labor and expense involved also favor the 96-well plate format. There was some loss of discovered phages using this technique, primarily targeting bacterial species less prevalent in the environment. This is an easily modifiable method that is amenable to automation and a variety of potential phage sources.

## Introduction

After the discovery of bacteriophages in the early 1900s, there was early success in the treatment of some bacterial infections [1–3]. However, the advent of small molecule antibiotics superseded their use in most countries as an antibiotic therapy. It was the growth of antimicrobial resistant infections and the absence of new options in the antibiotics discovery pipeline that led to renewed interest in bacteriophages as a therapeutic tool [4]. A highly popularized use of personalized phage therapy was in the treatment of a multidrug-resistant *Acinetobacter baumannii* infection in 2016 [5]. Numerous case reports and case series are now available in the literature demonstrating the successes and challenges of clinical phage therapy [6–8].

With the growth of bacteriophage therapy, the need for diverse phage banks and repositories has increased substantially [9, 10]. The ancient predator-prey relationship between phages

that includes information used to create figures and tables. This data set contains three main excel files: i) Cost comparison; ii) Growth curves to demonstrate identification of phage from waste water; and iii) High throughput phage discovery. This documentation is in addition to information already available in the manuscript, tables and figures.

**Funding:** This study was supported in part by the Dr. Louis Skarlow Memorial Trust award (Award number 94798) to OGG-D. The funder was not involved in study design, data collection and analysis, decision to publish, or preparation of the manuscript. There is no website for the foundation - relevant data including Form 990s are publicly available at https://www.causeiq.com/organizations/louis-sklarow-memorial-fund-836720,166201243/

**Competing interests:** The authors have declared that no competing interests exist.

and their hosts led to a variety of defense mechanisms, both active and passive [11] that allow most phages in nature to infect only a small proportion of bacteria in a given genus or species [12, 13], a phenomenon also called "host range" of a phage. In addition, even if a phage can initially infect a clinical isolate, the risk of subsequent development of resistance to that phage during a course of treatment is quite high. Finally, though phages are typically safe in their administration, the immune system recognizes them as non-self and can develop neutralizing antibodies against a particular phage [14, 15]. These factors lead to an inescapable conclusion–for a clinical phage program to treat most or all prospective patients, a number of phages for each possible host need to be available.

Phage sources are typically environmental due to the diversity of species encountered [13, 16]. Methods used to isolate phages from these sources are quite varied. These include, but are not limited to, simply spot testing a sample on a plaque assay, ultracentrifugation techniques, and enrichment (Fig 1) [17–21]. Each of these has multiple variations depending on the source of prospective phage material, the host organism, and growth constraints. Universal in these methods is the labor involved, particularly with ultracentrifugation and enrichment. Handling is required at multiple steps, and the higher yield techniques are time consuming as well. It can take 3–4 days or more from start to finish, with no promise of success until the plaque assay or other growth inhibition testing demonstrates the presence of phage. For anaerobic organisms or mycobacteria, the time to detection of a phage can be much longer [20]. In this study, we recognize the importance of regular availability of lytic phages in the treatment of resistant bacterial pathogens. Accordingly, the goal of this study is to evaluate a new low-volume enrichment method for rapid screening and isolation of lytic phages from environmental sources that has automation potential, is not labor intensive, and is faster than conventional methods [22].

## Methods

### Bacterial strains

Clinical bacterial isolates from Kaleida Health Clinical Laboratory (Buffalo, NY) were used in this study. All strains were stored at -80˚C prior to use (Table 1). All work with clinical bacterial isolates was performed in a biosafety level 2 (BSL-2) laboratory setting.

### Wastewater preparation

In April 2023, a local wastewater treatment facility (Amherst, NY) provided wastewater as dilute influent (DI) and return activated sludge (RAS). Four samples in total were obtained: Two one-liter samples were from "dilute influent" and two one-liter samples from "return activated sludge." Dilute influent (DI) refers to wastewater as it enters the treatment facility. Return activated sludge (RAS) is a concentrated bioactive material that represents wastewater that is heavily concentrated with a higher bioburden and presence of solid material prior to sterilization. DI wastewater is more aerobic, while RAS is a nearly anaerobic environment. These were kept separate throughout the experiment. Wastewater was centrifuged and serially filtered with final passage through sterile 0.2μm filters.

### Bacteriophage enrichment method comparison

To determine the utility of the low-volume enrichment method and compare it to more traditional enrichment methods for detection of bacteriophages, a comparison study was designed. Using the 4 waste sources (RAS #1, RAS #2, DI #1, and DI #2) and 66 clinical bacterial isolates, the low-volume and traditional enrichment methods were performed on each condition and

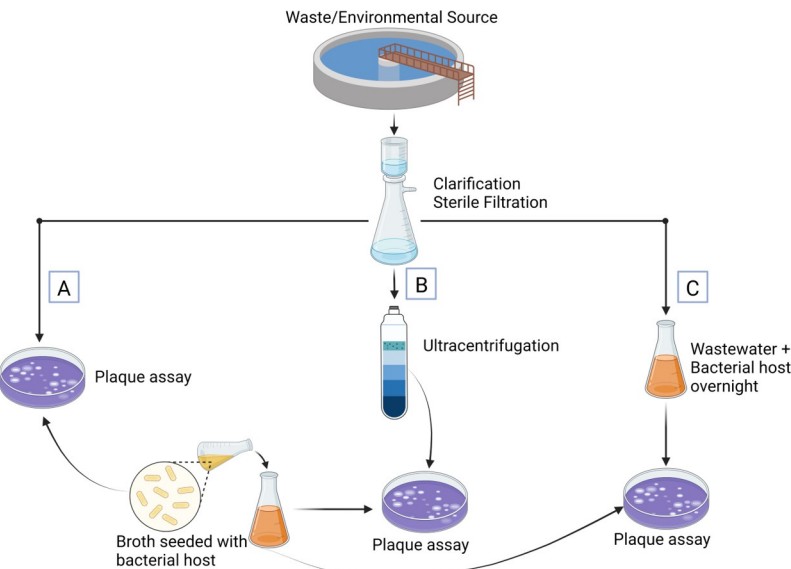

**Fig 1. Traditional phage isolation methods.** (A) Double agar plaque assay for isolating high titer phages. (B) Ultracentrifugation, typically with cesium chloride gradient, allows for concentration of phage prior to spot testing. (C) Enrichment methods select for phages in a sample that target the paired bacterial host".

isolate. Aerobic conditions and only a single media type (Luria-Bertani broth, LB) were used to better replicate conditions amenable to reduced hands-on time and eventual automation.

In the traditional bacteriophage enrichment method, 3mL wastewater and an equal volume of double concentration Lennox LB broth (RPI, Mt. Prospect, IL) with 60μL of overnight growth of the bacterial host was grown overnight at 37˚C and 170 RPM in an orbital shaker/incubator (Orbit Environ-shaker, Lab-Line, Melrose Park, IL)(Fig 1C, for reference). The product was centrifuged, sterile filtered, and spot tested via plaque assay against the homologous bacterial strain. Samples with phage were frozen at -80˚C for future use.

In the low-volume bacteriophage enrichment method, sterile 96-well plates were prepared (Fig 2). Control wells for each prospective phage host were loaded with 200 μL LB broth and 2–10 μL of overnight growth of the prospective bacterial host. Test wells were loaded using 100 μL of filtered wastewater, 100μL of a double concentration LB broth, and 2–10μL of the host. Prepared 96-well plates were placed in an Accuris Smartreader 9600T (Benchmark Scientific, Edison, NJ) for 18–24 hours at 37˚C and one-dimensional shaking at 4.7Hz. Readings of absorbance at 600nm were taken every 15 minutes. Data was analyzed with Microsoft Excel software. Wells with evidence of growth inhibition were sterile filtered and spot tested via plaque assay. Samples with phage were frozen at -80˚C for future use.

## Plaque assays

Plaque assays were performed to test for the presence of phage by pouring a mixture of 1mL overnight bacterial host with 4mL molten 0.7% LB agar (RPI, Mt. Prospect, IL) over a plate of LB agar in a standard 100mm disposable petri dish. The mixture for testing was applied to the plate in 5–10μL drops. These were allowed to dry and the plate was then placed in an incubator overnight at 37˚C.

**Table 1. Bacterial clinical isolates tested as possible phage hosts.** *Campylobacter* and *Lactobacillus* species were not tested for further identification. *Shigella* listed as species were from serotype A. Abbreviations: ESBL: extended spectrum beta lactamase; ICU: Intensive Care Unit; spp: species; XDR: extensively drug resistant.

| Genus | Species | Isolate identifier | Source |
|---|---|---|---|
| *Achromobacter* | *xylosoxidans* | achxylcf | Adult, ICU, blood, ESBL, aminoglycoside-resistant |
| *Acinetobacter* | *baumannii* | acibaumc1 | Pediatric, ICU, tracheal aspirate, susceptible isolate |
| *Acinetobacter* | *baumannii* | HUMC1 | Adult, XDR |
| *Acinetobacter* | *baumannii* | acibaues | Pediatric, oncology ward, blood, ESBL |
| *Acinetobacter* | *baumannii* | HUMC12 | Adult, XDR |
| *Acinetobacter* | *baumannii* | Kelly Lung | Adult, tracheal aspirate, carbapenem-resistant |
| *Acinetobacter* | *baumannii* | acibaumc2 | Pediatric, ICU, tracheal aspirate, ESBL |
| *Acinetobacter* | *baumannii* | RPCI AB1 | Adult, oncology ward, XDR |
| *Acinetobacter* | *baumannii* | acibaucf | Adult, ICU, tracheal aspirate, susceptible isolate |
| *Acinetobacter* | *baumannii* | 307 | Adult, ESBL |
| *Acinetobacter* | *haemolyticus* | HR1 | Adult, meropenem resistant |
| *Acinetobacter* | *ursingii* | aciurses | Pediatric, oncology ward, blood, ESBL |
| *Bordetella* | *pertussis* | BSA SPG2 | Pediatric, sputum |
| *Bordetella* | *pertussis* | WT1 | Pediatric, sputum |
| *Bordetella* | *pertussis* | Tahoma | Pediatric, sputum |
| *Campylobacter* | spp | 2TS | Pediatric, stool |
| *Campylobacter* | spp | 3TS | Pediatric, stool |
| *Citrobacter* | *amalonaticus* | 02451 | Pediatric, blood, ESBL |
| *Enterobacter* | *cloacae* | entclodc1 | Adult, ICU, infected peritoneal fluid, ESBL |
| *Enterobacter* | *cloacae* | entclodc2 | Adult, ICU, abdominal abscess, carbapenem-resistant |
| *Enterobacter* | *cloacae* | entclodc3 | Adult, ICU, tracheal aspirate, susceptible only to cefiderocol/amikacin |
| *Escherichia* | *coli* | ecosq | Adult, ICU |
| *Escherichia* | *coli* (O157:H7) | 08063 | Pediatric, stool |
| *Escherichia* | *coli* | ecomw | Adult, ICU, blood, ESBL |
| *Escherichia* | *coli* | CSF | Pediatric, meningitis |
| *Klebsiella* | *pneumoniae* | stock | Adult, blood, ESBL |
| *Klebsiella* | *pneumoniae* | klepnakp | Adult, ICU, tracheal aspirate, carbapenem-resistant |
| *Klebsiella* | *pneumoniae* | ATCC BAA1705 | Adult, urine, carbapenem-resistant (KPC) |
| *Lactobacillus* | spp | 2181–069 | Pediatric, stool |
| *Lactobacillus* | spp | 2181–080 | Pediatric, stool |
| *Lactobacillus* | spp | 2181–044 | Pediatric, stool |
| *Lactobacillus* | spp | 2181–005 | Pediatric, stool |
| *Lactobacillus* | spp | 2181–045 | Pediatric, stool |
| *Lactobacillus* | spp | 2181–028 | Pediatric, stool |
| *Lactobacillus* | spp | 2181–017 | Pediatric, stool |
| *Lactobacillus* | spp | 2198–088 | Pediatric, stool |
| *Proteus* | *mirabilis* | 2161–017 | Pediatric, urine |
| *Proteus* | *mirabilis* | 2161–047 | Pediatric, urine |
| *Proteus* | *mirabilis* | 2161–021 | Pediatric, urine |
| *Proteus* | *mirabilis* | 2161–019 | Pediatric, urine |
| *Pseudomonas* | *aeruginosa* | 02451 | Pediatric, blood |
| *Pseudomonas* | *aeruginosa* | psarma2 | Pediatric, tracheal aspirate, ESBL |
| *Pseudomonas* | *aeruginosa* | psaerw | Adult, ICU, abdominal drainage, susceptible to cefiderocol alone |
| *Pseudomonas* | *aeruginosa* | psarma1 | Pediatric, tracheal aspirate, XDR |
| *Salmonella* | *enterica* Type B | B 1942–071 | Pediatric, stool |
| *Salmonella* | *enterica* Type B | 1999–002 | Pediatric, stool |

*(Continued)*

**Table 1.** (Continued)

| Genus | Species | Isolate identifier | Source |
|---|---|---|---|
| *Salmonella* | *enterica* Type B | stock | Pediatric, stool |
| *Salmonella* | *typhi* | 03802 | Pediatric, stool |
| *Salmonella* | *typhimurium* | SL3261 | Pediatric, stool |
| *Shigella* | *boydii* | SH42 | Pediatric, stool |
| *Shigella* | *flexneri* | SH24 | Pediatric, stool |
| *Shigella* | *sonnei* | SH22 | Pediatric, stool |
| *Shigella* | spp | shigjs | Adult, stool, ESBL and quinolone-resistant |
| *Shigella* | spp | SH18 | Pediatric, stool |
| *Shigella* | spp | SH16 | Pediatric, stool |
| *Staphylococcus* | *aureus* | MRSA 00175 | Pediatric, blood, methicillin-resistant |
| *Staphylococcus* | *aureus* | MRSA 14161 | Pediatric, blood, methicillin-resistant |
| *Staphylococcus* | *aureus* | MSSA 00175 | Pediatric, blood, methicillin-sensitive |
| *Staphylococcus* | *aureus* | MRSA 06970 | Pediatric, blood, methicillin-resistant |
| *Staphylococcus* | coagulase negative | 00751 | Pediatric, blood, methicillin-resistant |
| *Stenotrophomonas* | *maltophilia* | stemalma | Pediatric, tracheal aspirate |
| *Stenotrophomonas* | *maltophilia* | stematl2 | Pediatric, tracheal aspirate, quinolone-resistant |
| *Stenotrophomonas* | *maltophilia* | stematl1 | Pediatric, tracheal aspirate |
| *Vibrio* | *cholerae* | El Tor N16961 | Pediatric, stool |
| *Yersinia* | *enterocolitica* | 1992–045 | Pediatric, stool |
| *Yersinia* | *enterocolitica* | 1999–034 | Pediatric, stool |

## Data analysis

Numeric data from all experimental work were entered in Microsoft Excel datasheets available at URL: DOI 10.17605/OSF.IO/AHGJF. Data from traditional-enrichment versus low-volume enrichment methods were compared with software using student's T at 95% confidence. Variables included experimental hands-on time, consumable costs, and phage recovery as a function of wastewater source and method at the isolate and species levels (**S1–S3 Files**).

## Ethics statement

This study involved the use of bacterial clinical isolates from the Kaleida Health Clinical Laboratory (Buffalo, NY). We obtained University at Buffalo Institutional Review Board (IRB) approval under MODCR00007172 protocol to use deidentified clinical bacterial isolates for this study. No participants were recruited and no prospective or retrospective clinical information was collected as part of this study. A request for waiver of consent was approved by the IRB. Accordingly, no consent was obtained from human subjects for the use of bacterial isolates in this study. A total of 66 deidentified clinical isolates were evaluated in this study.

The bacteriophage research plan, including collection of water samples as well as detection, isolation, purification, and characterization of phages, was exempted under IRB STUDY00002299.

## Results

### Time and cost benefit with low-volume bacteriophage enrichment method

The "hands-on" time, total experimental time, and monetary costs of the methods were calculated. As the initial wastewater processing and bacterial culture starts were similar between the methods, these were not evaluated. Time for each subsequent step was monitored and

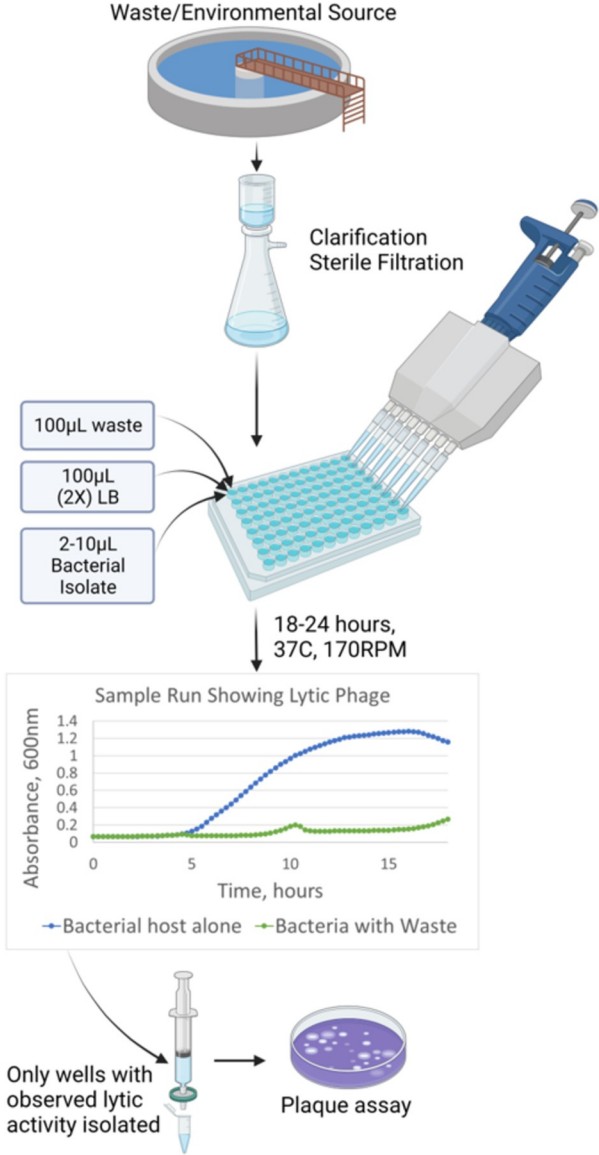

**Fig 2. Dynamic bacterial growth with the low-volume bacteriophage enrichment method.** Wastewater is clarified and sterile filtered. It is then combined with an equal volume of double concentration (2X) Luria-Bertani (LB) broth and overnight growth of the bacterial host. Wells with probable lytic phage are tested by double agar plaque assay".

averaged to create a comparison table (Table 2). This demonstrated a >50% reduction in hands-on time with the low-volume method (7.5 vs 18.7 hours). Phage lysis in the low-volume method was seen in all positive isolates less than 10 hours after plates were started. Plaque visualization by the traditional method required a minimum of 48 hours after enrichment was started. For *Bordetella* and *Lactobacillus* isolates, plaque assays were unreadable until at least 72–96 hours from enrichment initiation due to delayed growth. Consumable costs were also evaluated (Table 3). The low-volume method was significantly less expensive for the experiment ($206 vs $558). If traditional methods had been used to confirm all "negative" results in the low-volume method (i.e. the absence of lytic activity), the hands-on time and monetary

**Table 2. Hands-on time comparison between low-volume and traditional phage enrichment methods.** Plate setup refers to the time required from obtaining supplies to starting the plate reader. Enrichment setup refers to the time required to assemble each tube used in the traditional enrichment method.

| Item | Low-volume | | | Traditional | | |
|------|--------|---------------------|----------------|--------|---------------------|----------------|
| | Number | Time (each, minutes) | Time (minutes) | Number | Time (each, minutes) | Time (minutes) |
| Plate setup | 8 | 30 | 240 | - | - | - |
| Plate data evaluation | 8 | 5 | 40 | - | - | - |
| Enrichment setup | - | - | - | 264 | 2 | 528 |
| Filtration | 75 | 0.5 | 37.5 | 264 | 1 | 264 |
| Plaque assays | 26 | 5 | 130 | 66 | 5 | 330 |
| | | Total (minutes): | 447.5 | | Total (minutes): | 1122 |
| | | Total (hours): | 7.5 | | Total (hours): | 18.7 |

costs would have been nearly equal to the traditional method alone at 18.8 hours and $550, respectively, without any increase in total time to phage isolation.

## Phage recovery comparison between low-volume and traditional bacteriophage enrichment methods

The four wastewater samples were tested in each method in combination with 66 clinical isolates. Overall, phages were isolated that targeted 43 of 66 (65%) clinical isolates (Table 4). Phages targeting 26 (39%) of 66 isolates were isolated utilizing the low-volume 96-well plate method, while 43 (65%) isolates had lytic phage discovered with traditional enrichment. This difference was significant (p = 0.0028) at 95% confidence. Coagulase-negative *Staphylococcus* (non-*aureus*), *Vibrio cholerae*, and *Yersinia enterocolitica* were the only species without plaques found in conventional plaque assays. Phages targeting 13 (54%) species were isolated with the low-volume 96-well method compared to 21 (88%) with traditional enrichment (p = 0.011).

## Growth limitations regardless of enrichment method

The use of aerobic conditions and only a single media did limit growth of some organisms, specifically *B. pertussis* and *Lactobacillus* species. The absence of growth correlated with both the isolation method and wastewater source. 11 of 66 (16.7%) isolates in the RAS/low-volume groups did not grow compared to 6 (9.1%) isolates in the DI/low-volume groups. Five of 66

**Table 3. Monetary cost comparison between low-volume and traditional phage enrichment methods.** Costs, itemized as much as possible, were based on local stockroom costs at the time this manuscript was written. Cost variations are expected among institutions and over time.

| Item | Cost | #, microplate | Cost, microplate | #, traditional | Cost, traditional |
|------|------|---------------|------------------|----------------|-------------------|
| 96-well plates | $2.08 each | 8 | $ 17 | 0 | $ - |
| Culture tubes | $6.91 per 25 | 66 | $ 18 | 330 | $ 91 |
| LB broth (mL) | $58.88 per 500g mix | 286 | $ 1 | 1848 | $ 5 |
| LB agar (mL) | $93 per 500g mix | 416 | $ 3 | 1056 | $ 8 |
| Petri dishes | $2.98 per 25 | 26 | $ 3 | 66 | $ 8 |
| Syringe filters, 0.22μm | $66.12 per 50 | 75 | $ 99 | 264 | $ 349 |
| 1mL syringe | $38.17 per 200 | 75 | $ 14 | 264 | $ 50 |
| 20μL pipette tips (No. of boxes) | $5.38 per box | 4 | $ 22 | 4 | $ 22 |
| 200μL pipette tips (No. of boxes) | $5.38 per box | 2 | $ 11 | 1 | $ 5 |
| 5mL pipettes | $9.20 per 50 | 99 | $ 18 | 157 | $ 29 |
| | | Total: | $ 206 | Total: | $ 568 |

**Table 4. Results of enrichment product testing for the presence of phage.** "+" indicates lytic phage was isolated. "-" indicates no phage was detected. Abbreviations: DI: Dilute Influent; RAS: Return Activated Sludge.

| Genus | Species | Isolate identifier | RAS 1 | | RAS 2 | | DI 1 | | DI 2 | |
|---|---|---|---|---|---|---|---|---|---|---|
| | | | 96-well | Traditional | 96-well | Traditional | 96-well | Traditional | 96-well | Traditional |
| *Achromobacter* | *xylosoxidans* | achxylcf | - | + | - | - | - | - | - | + |
| *Acinetobacter* | *baumannii* | acibaumc1 | - | - | - | - | - | - | - | - |
| *Acinetobacter* | *baumannii* | HUMC1 | - | - | - | - | - | - | - | + |
| *Acinetobacter* | *baumannii* | acibaues | - | + | - | + | - | - | - | - |
| *Acinetobacter* | *baumannii* | HUMC12 | - | + | - | + | - | + | - | + |
| *Acinetobacter* | *baumannii* | Kelly Lung | - | - | - | - | - | - | - | - |
| *Acinetobacter* | *baumannii* | acibaumc2 | - | - | - | - | - | - | - | - |
| *Acinetobacter* | *baumannii* | RPCI AB1 | - | + | - | + | - | - | - | - |
| *Acinetobacter* | *baumannii* | acibaucf | - | - | - | - | - | - | - | - |
| *Acinetobacter* | *baumannii* | 307 | - | + | - | + | - | + | - | + |
| *Acinetobacter* | *haemolyticus* | HR1 | - | + | - | - | - | + | - | + |
| *Acinetobacter* | *ursingii* | aciurses | - | + | - | + | - | + | - | + |
| *Bordetella* | *pertussis* | BSA SPG2 | - | + | - | + | + | - | + | - |
| *Bordetella* | *pertussis* | WT1 | - | - | - | - | - | - | - | - |
| *Bordetella* | *pertussis* | Tahoma | - | + | - | + | + | - | + | - |
| *Campylobacter* | spp | 2TS | + | + | + | + | + | + | + | + |
| *Campylobacter* | spp | 3TS | + | + | + | + | + | + | + | + |
| *Citrobacter* | *amalonaticus* | 02451 | - | + | - | - | - | + | - | + |
| *Enterobacter* | *cloacae* | entclodc1 | + | + | + | + | - | + | - | + |
| *Enterobacter* | *cloacae* | entclodc2 | + | + | + | + | - | + | + | + |
| *Enterobacter* | *cloacae* | entclodc3 | + | + | + | + | - | - | - | + |
| *Escherichia* | *coli* | ecosq | + | + | + | + | + | + | + | + |
| *Escherichia* | *coli* (O157:H7) | 08063 | + | + | + | + | + | + | + | + |
| *Escherichia* | *coli* | ecomw | - | - | - | + | - | + | - | + |
| *Escherichia* | *coli* | CSF | + | + | + | + | + | + | + | + |
| *Klebsiella* | *pneumoniae* | stock | - | + | + | + | - | + | + | + |
| *Klebsiella* | *pneumoniae* | klepnakp | - | - | - | - | - | - | - | - |
| *Klebsiella* | *pneumoniae* | ATCC BAA1705 | + | + | + | + | + | + | + | + |
| *Lactobacillus* | spp | 2181–069 | - | - | - | - | - | - | - | - |
| *Lactobacillus* | spp | 2181–080 | - | - | - | - | - | - | - | - |
| *Lactobacillus* | spp | 2181–044 | - | - | - | - | - | - | - | - |
| *Lactobacillus* | spp | 2181–005 | - | - | - | - | - | - | - | - |
| *Lactobacillus* | spp | 2181–045 | - | - | - | + | - | - | - | + |
| *Lactobacillus* | spp | 2181–028 | - | - | - | - | - | - | - | - |
| *Lactobacillus* | spp | 2181–017 | - | - | - | - | - | - | - | - |
| *Lactobacillus* | spp | 2198–088 | - | - | - | - | - | - | - | - |
| *Proteus* | *mirabilis* | 2161–017 | - | - | - | - | - | - | - | - |
| *Proteus* | *mirabilis* | 2161–047 | - | + | - | - | - | - | - | - |
| *Proteus* | *mirabilis* | 2161–021 | - | - | - | - | - | - | - | - |
| *Proteus* | *mirabilis* | 2161–019 | - | - | - | - | - | - | - | - |
| *Pseudomonas* | *aeruginosa* | 02451 | - | + | - | - | - | + | - | + |
| *Pseudomonas* | *aeruginosa* | psarma2 | + | + | + | + | - | + | - | + |
| *Pseudomonas* | *aeruginosa* | psaerw | + | + | - | + | - | + | - | + |
| *Pseudomonas* | *aeruginosa* | psarma1 | - | + | - | - | - | + | - | - |
| *Salmonella* | *enterica* Type B | B 1942–071 | - | - | - | - | - | + | - | + |

(*Continued*)

**Table 4.** (Continued)

| Genus | Species | Isolate identifier | RAS 1 | | RAS 2 | | DI 1 | | DI 2 | |
|---|---|---|---|---|---|---|---|---|---|---|
| | | | 96-well | Traditional | 96-well | Traditional | 96-well | Traditional | 96-well | Traditional |
| *Salmonella* | *enterica* Type B | 1999–002 | - | - | - | + | - | - | - | - |
| *Salmonella* | *enterica* Type B | stock | + | + | + | + | + | + | + | + |
| *Salmonella* | *typhi* | 03802 | + | + | - | + | - | + | + | + |
| *Salmonella* | *typhimurium* | SL3261 | - | - | - | - | - | + | + | + |
| *Shigella* | *boydii* | SH42 | + | + | + | + | - | + | - | + |
| *Shigella* | *flexneri* | SH24 | + | + | + | + | + | + | + | + |
| *Shigella* | *sonnei* | SH22 | + | + | + | + | + | + | + | + |
| *Shigella* | spp | shigjs | + | + | + | + | + | + | + | + |
| *Shigella* | spp | SH18 | + | + | + | + | + | + | + | + |
| *Shigella* | spp | SH16 | + | + | + | + | + | - | + | - |
| *Staphylococcus* | *aureus* | MRSA 00175 | - | - | - | - | - | - | - | - |
| *Staphylococcus* | *aureus* | MRSA 14161 | - | - | - | - | - | + | - | + |
| *Staphylococcus* | *aureus* | MSSA 00175 | - | - | - | - | - | - | - | - |
| *Staphylococcus* | *aureus* | MRSA 06970 | - | - | - | - | - | - | - | - |
| *Staphylococcus* | coagulase negative | 00751 | - | - | - | - | - | - | - | - |
| *Stenotrophomonas* | *maltophilia* | stemalma | - | + | - | + | + | + | + | + |
| *Stenotrophomonas* | *maltophilia* | stematl2 | - | - | + | - | - | - | - | - |
| *Stenotrophomonas* | *maltophilia* | stematl1 | + | + | + | + | - | + | + | + |
| *Vibrio* | *cholerae* | El Tor N16961 | - | - | - | - | - | - | - | - |
| *Yersinia* | *enterocolitica* | 1992–045 | - | - | - | - | - | - | - | - |
| *Yersinia* | *enterocolitica* | 1999–034 | - | - | - | - | - | - | - | - |

(7.6%) isolates did not grow in the traditional enrichment method independent of the wastewater source. It may be posited that the larger volume of the traditional enrichment method allowed for a more anaerobic environment that the microplates, favoring the growth of *Lactobacillus*.

## Phage isolation concordance between methods

Concordance between the 2 methods was present in 44–52 (67–79%) of the 66 isolates tested when all 4 wastewater samples were evaluated (Table 5). Concordance fell slightly at the species level. 14–19 (58–79%) of the 24 species tested were concordant between methods.

## Discussion

In this study, phages targeting a wide variety of clinically relevant pathogens were recovered from a community wastewater treatment facility. This increases the number of phage candidates for use in therapy, but also increases the general knowledge of phage binding and activity.

Neither method was successful in isolating phages targeting coagulase-negative *Staphylococci*, *V. cholerae*, or *Y. enterocolitica* in this study. Compared to traditional enrichment, the low-volume method was unsuccessful in isolating phages targeting *Achromobacter xylosoxidans*, *Acinetobacter baumannii*, *Acinetobacter haemolyticus*, *Acinetobacter ursingii*, *Citrobacter amalonaticus*, *Lactobacillus* spp, *Proteus mirabilis*, or *Staphylococcus aureus*. This suggests that these phages were less prevalent in the wastewater, requiring a higher volume to allow for enrichment. The overall phage recovery was increased in RAS compared to DI. RAS is

**Table 5. Congruence of phage isolation results with individual bacterial isolates.** A comparison between low-volume and traditional enrichment methods is demonstrated using a heat map. Green boxes represent samples in which the 2 methods had congruent results. Orange boxes represent samples in which the 2 methods had discongruent results. Typically this favored phage isolation with the traditional enrichment method, though with *Bordetella pertussis* the low volume format was favored. Abbreviations: RAS: Return Activated Sludge; spp: species; MRSA: methicillin-resistant *Staphylococcus aureus*.

| Host Genus | Species | Isolate identifier | RAS 1 | RAS 2 | DI 1 | DI 2 |
|---|---|---|---|---|---|---|
| *Achromobacter* | *xylosoxidans* | achxylcf | 0 | 1 | 1 | 0 |
| *Acinetobacter* | *baumannii* | acibaumc1 | 1 | 1 | 1 | 1 |
| *Acinetobacter* | *baumannii* | HUMC1 | 1 | 1 | 1 | 0 |
| *Acinetobacter* | *baumannii* | acibaues | 0 | 0 | 1 | 1 |
| *Acinetobacter* | *baumannii* | HUMC12 | 0 | 0 | 0 | 0 |
| *Acinetobacter* | *baumannii* | Kelly Lung | 1 | 1 | 1 | 1 |
| *Acinetobacter* | *baumannii* | acibaumc2 | 1 | 1 | 1 | 1 |
| *Acinetobacter* | *baumannii* | RPCI AB1 | 0 | 0 | 1 | 1 |
| *Acinetobacter* | *baumannii* | acibaucf | 1 | 1 | 1 | 1 |
| *Acinetobacter* | *baumannii* | 307 | 0 | 0 | 0 | 0 |
| *Acinetobacter* | *haemolyticus* | HR1 | 0 | 1 | 0 | 0 |
| *Acinetobacter* | *ursingii* | aciurses | 0 | 0 | 0 | 0 |
| *Bordetella* | *pertussis* | BSA SPG2 | 0 | 0 | 0 | 0 |
| *Bordetella* | *pertussis* | WT1 | 0 | 0 | 0 | 0 |
| *Bordetella* | *pertussis* | Tahoma | 0 | 0 | 0 | 0 |
| *Campylobacter* | spp | 2TS | 1 | 1 | 1 | 1 |
| *Campylobacter* | spp | 3TS | 1 | 1 | 1 | 1 |
| *Citrobacter* | *amalonaticus* | 02451 | 0 | 1 | 0 | 0 |
| *Enterobacter* | *cloacae* | entclodc1 | 1 | 1 | 0 | 0 |
| *Enterobacter* | *cloacae* | entclodc2 | 1 | 1 | 0 | 1 |
| *Enterobacter* | *cloacae* | entclodc3 | 1 | 1 | 1 | 0 |
| *Escherichia* | *coli* | ecosq | 1 | 1 | 1 | 1 |
| *Escherichia* | *coli* | 08063 (O157:H7) | 1 | 1 | 1 | 1 |
| *Escherichia* | *coli* | ecomw | 1 | 0 | 0 | 0 |
| *Escherichia* | *coli* | CSF | 1 | 1 | 1 | 1 |
| *Klebsiella* | *pneumoniae* | stock | 0 | 1 | 0 | 1 |
| *Klebsiella* | *pneumoniae* | klepnakp | 1 | 1 | 1 | 1 |
| *Klebsiella* | *pneumoniae* | ATCC RAA1705 | 1 | 1 | 1 | 1 |
| *Lactobacillus* | spp | 2181–069 | 1 | 1 | 1 | 1 |
| *Lactobacillus* | spp | 2181–080 | 0 | 0 | 1 | 1 |
| *Lactobacillus* | spp | 2181–044 | 1 | 1 | 1 | 1 |
| *Lactobacillus* | spp | 2181–005 | 1 | 1 | 1 | 1 |
| *Lactobacillus* | spp | 2181–045 | 0 | 0 | 1 | 0 |
| *Lactobacillus* | spp | 2181–028 | 0 | 0 | 0 | 0 |
| *Lactobacillus* | spp | 2181–017 | 1 | 1 | 0 | 0 |
| *Lactobacillus* | spp | 2198–088 | 1 | 1 | 1 | 1 |
| *Proteus* | *mirabilis* | 2161–017 | 1 | 1 | 1 | 1 |
| *Proteus* | *mirabilis* | 2161–047 | 0 | 1 | 1 | 1 |
| *Proteus* | *mirabilis* | 2161–021 | 1 | 1 | 1 | 1 |
| *Proteus* | *mirabilis* | 2161–019 | 1 | 1 | 1 | 1 |
| *Pseudomonas* | *aeruginosa* | 02451 | 0 | 1 | 0 | 0 |
| *Pseudomonas* | *aeruginosa* | psarma2 | 1 | 1 | 0 | 0 |
| *Pseudomonas* | *aeruginosa* | psaerw | 1 | 0 | 0 | 0 |
| *Pseudomonas* | *aeruginosa* | psarma1 | 0 | 1 | 0 | 1 |
| *Salmonella* | *enterica* | B 1942–071 | 1 | 1 | 0 | 0 |

*(Continued)*

**Table 5.** (Continued)

| Host Genus | Species | Isolate identifier | RAS 1 | RAS 2 | DI 1 | DI 2 |
|---|---|---|---|---|---|---|
| *Salmonella* | *enterica B* | 1999–002 | 1 | 0 | 1 | 1 |
| *Salmonella* | *enterica Type B* | stock | 1 | 1 | 1 | 1 |
| *Salmonella* | *typhi* | 03802 | 1 | 0 | 0 | 1 |
| *Salmonella* | *typhimurium* | SL3261 | 1 | 1 | 0 | 1 |
| *Shigella* | *boydii* | SH42 | 1 | 1 | 0 | 0 |
| *Shigella* | *flexneri* | SH24 | 1 | 1 | 1 | 1 |
| *Shigella* | *sonnei* | SH22 | 1 | 1 | 1 | 1 |
| *Shigella* | *species* | shigjs | 1 | 1 | 1 | 1 |
| *Shigella* | spp | SH18 | 1 | 1 | 1 | 1 |
| *Shigella* | spp | SH16 | 1 | 1 | 0 | 0 |
| *Staphylococcus* | *aureus* | MRSA 00175 | 1 | 1 | 1 | 1 |
| *Staphylococcus* | *aureus* | MRSA 14161 | 1 | 1 | 0 | 0 |
| *Staphylococcus* | *aureus* | MSSA 00175 | 1 | 1 | 1 | 1 |
| *Staphylococcus* | *aureus* | MRSA 06970 | 1 | 1 | 1 | 1 |
| *Staphylococcus* | *coagulase negative* | 00751 | 1 | 1 | 1 | 1 |
| *Stenotrophomonas* | *maltophilia* | stemalma | 0 | 0 | 1 | 1 |
| *Stenotrophomonas* | *maltophilia* | stematl2 | 1 | 0 | 1 | 1 |
| *Stenotrophomonas* | *maltophilia* | stematl1 | 1 | 1 | 0 | 1 |
| *Vibrio* | *cholerae* | El Tor N16961 | 1 | 1 | 1 | 1 |
| *Yersinia* | *enterocolitica* | 1992–045 | 1 | 1 | 1 | 1 |
| *Yersinia* | *enterocolitica* | 1999–034 | 1 | 1 | 1 | 1 |

returned biomaterial from the last step before sterilization, and is overall a rich sample in a more anaerobic environment than DI. The species against which phage was recovered did not significantly differ between RAS and DI sources with the exception of anti-Acinetobacter and anti-MRSA phages which were only recovered in the more aerobic environment of DI.

In order to create conditions that were more readily amenable to automation and increase the ease of assay setup, we used only a single culture media (LB). *Lactobacillus* and *Bordetella*, organisms with anaerobic and special media requirements, respectively, were specifically included in the data set in order to have some representation of difficult to grow organisms. All 3 *Bordetella* isolates grew in the higher volume traditional method, while only 2 grew in the low-volume method. Only 3 of the 8 *Lactobacillus* isolates grew in each method. It may be posited that the larger volume of the traditional enrichment method allowed for a more anaerobic environment than the microplates, favoring the growth of Lactobacillus. If these genera are removed from the data set, concordance between the methods is not significantly affected, changing from 73±8% to 72±10% (p = 0.87). However, at the level of individual isolates, percent recovery is positively affected for both data sets if *Lactobacillus* and *Bordetella* are removed. The low-volume enrichment method improves from 39% (26 of 66) to 44% recovery (24 of 55). The traditional enrichment method improves from 65% (43 of 66) to 73% (40 of 55).

Despite the limitations in isolating low titer phages and growing bacteria with special growth needs, phage recovery by the low-volume enrichment method was high, at 39% of individual isolates and 54% of tested species. Common *Enterobacteriaceae*-targeting phages were well represented, as were more common gram-negatives including *Pseudomonas* and *Stenotrophomonas*. The experimental time required to determine whether phage was present was comparatively quite short (Fig 3). All samples with phage were clearly delineated within 10 hours of starting the plate on day 2 (Fig 4). Only 3 of the 26 isolates with phage detected by low-

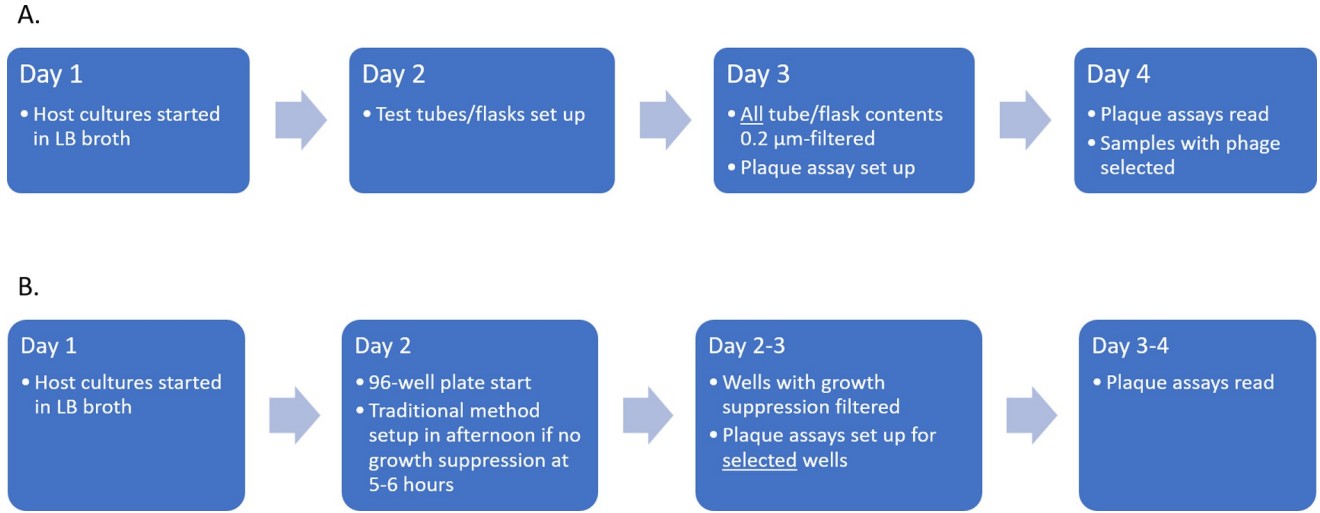

**Fig 3. Comparison of bacteriophage enrichment methods arranged by processing day.** (A) Traditional enrichment with phage identified on day 4 of testing. (B) Low volume enrichment with phage identified as early as day 2–3 of testing. Abbreviations: LB–Luria-Bertani".

volume enrichment took longer than 5 hours: these were slower growing isolates of *S. maltophilia*, *Campylobacter* spp., and *S. typhi*. All were subsequently confirmed by plaque assay. In contrast, only 3 of the traditional samples (all in *Shigella* species) had obvious clearing of culture suggestive of phage that was visible on day 3. The remainder required reading plaque assays on day 4. This offers a clear benefit to the low-volume method and a possible combinatorial approach where one could set up the plate in the morning of Day 2 and, approximately 5–6 hours later, start higher volume enrichment for samples with growth curves that do not

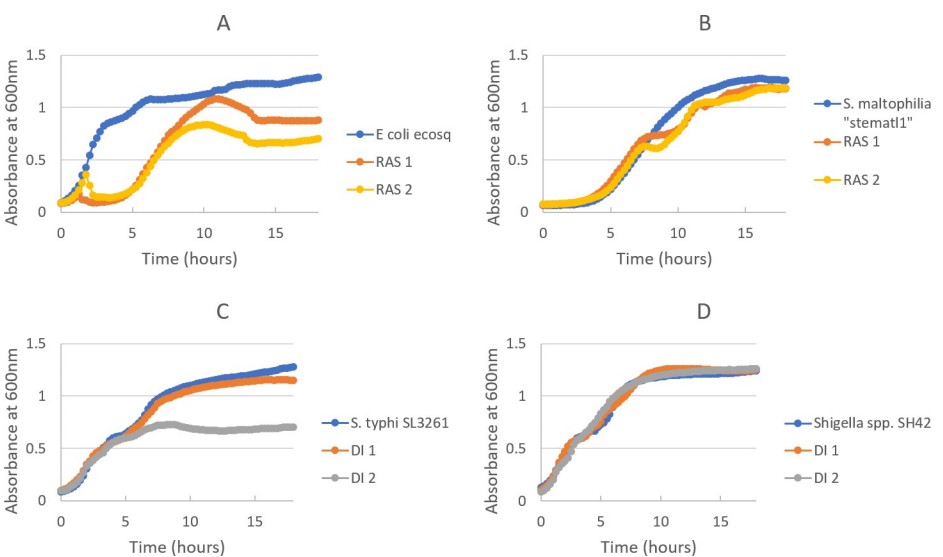

**Fig 4. Bacterial growth curves from low-volume enrichment method demonstrating waste water-mediated differential growth.** (A) *E. coli* "ecosq" grown alone and with RAS 1 and RAS 2, with phage isolated from both wastewater samples. (B) *S. maltophilia* "stematl1" grown alone and with RAS 1 and RAS 2, with phage isolated late from both wastewater samples. (C) *S. typhimurium* "SL3261" grown alone and with DI 1 and DI 2, with phage isolated only from DI 2. (D) *Shigella spp*. "SH42" grown alone and with DI 1 and DI 2, with no phage isolated".

demonstrate lytic activity. As plate set up only takes approximately 30 minutes, this would not be an overly long day and would not delay results from traditional enrichment.

A limitation of the low-volume method is a reduced chance of capturing low concentration phages. In these results, this included *Achromobacter*, *Acinetobacter*, *Proteus*, and *Staphylococcus* species. However, this data set also shows improved low-volume recovery of more common gram-negative species. While a broad phage search might require follow-up traditional enrichment or other isolation techniques for samples without phage capture in the low-volume method, most phage searches would be narrower in focus. In addition, concentration methods including ultrafiltration, tangential flow filtration, and ultracentrifugation could be used to increase the concentration of phage in an environmental sample. This is expected to improve the yield of the low-volume method.

There are other clear benefits for low-volume screening. The convenience of reduced overall experimental time to phage isolation is accompanied by reduced handling time and material costs. In traditional methods, all samples must be prepared equally. This includes, at a minimum, centrifugation for larger volumes, filtration of all samples, storage of all enriched cultures, and plaque assays for each individual host. The handling times and the costs of the involved consumables increase linearly with each additional isolate or wastewater sample tested. Reduction in these time and material costs are significant. An additional benefit in the low-volume screening method would include screening for patient samples in compassionate use scenarios. Time can be critical in patients with difficult-to-treat infections, and saving 2 days or more would be of substantial benefit.

## Conclusions

Traditional enrichment and low-volume enrichment methods were compared in this study evaluating 66 clinical isolates and 4 total waste samples representing the most commonly used waste types. Phage recovery was high overall with concordance between methods at approximately 73%. Concordance was higher in concentrated waste material. Overall time, hands-on time, and material costs were all significantly lower in the low-volume enrichment group, and the dynamic monitoring allows for early phage detection without the use of plaque assays. These results support the utility of low-volume screening as a viable method to reduce hands on time and overall time required for phage discovery, particularly with concentrated samples, screening a large number of isolates, or screening patient samples when time may be limited.

## Supporting information

**S1 File. Cost comparison table.**
(XLSX)

**S2 File. Growth curves to determine identification of phage from wastewater.**
(XLSX)

**S3 File. High throughput phage discovery master sheet.**
(XLSX)

## Acknowledgments

We would like to thank Dr. Karl O. A. Yu for providing helpful comments. We are grateful to Mr. John Richter and all personnel at the Amherst Wastewater Treatment Facility, Amherst, NY, for providing phage-containing wastewater samples.

## Author Contributions

**Conceptualization:** Patrick O. Kenney.

**Formal analysis:** Patrick O. Kenney.

**Funding acquisition:** Oscar G. Gómez-Duarte.

**Investigation:** Patrick O. Kenney, Oscar G. Gómez-Duarte.

**Methodology:** Patrick O. Kenney.

**Project administration:** Oscar G. Gómez-Duarte.

**Resources:** Oscar G. Gómez-Duarte.

**Supervision:** Oscar G. Gómez-Duarte.

**Writing – original draft:** Patrick O. Kenney, Oscar G. Gómez-Duarte.

**Writing – review & editing:** Patrick O. Kenney, Oscar G. Gómez-Duarte.

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
