## [Decision Letter · Decision Letter 0]

3 Jan 2024

PONE-D-23-39330Low-Volume Enrichment Method Supports High Throughput Bacteriophage Screening and Isolation from WastewaterPLOS ONE

Dear Dr. Gomez-Duarte,

Thank you for submitting your manuscript to PLOS ONE. After careful consideration, we feel that it has merit but does not fully meet PLOS ONE’s publication criteria as it currently stands. Therefore, we invite you to submit a revised version of the manuscript that addresses the points raised during the review process. Your manuscript has been reviewed by an expert in your field. A minor revision is suggested by the reviewer and I agree with the comments. Please submit your revised manuscript by Feb 17 2024 11:59PM. If you will need more time than this to complete your revisions, please reply to this message or contact the journal office at plosone@plos.org. Please include the following items when submitting your revised manuscript:A rebuttal letter that responds to each point raised by the academic editor and reviewer(s). You should upload this letter as a separate file labeled 'Response to Reviewers'.A marked-up copy of your manuscript that highlights changes made to the original version. You should upload this as a separate file labeled 'Revised Manuscript with Track Changes'.An unmarked version of your revised paper without tracked changes. You should upload this as a separate file labeled 'Manuscript'.If applicable, we recommend that you deposit your laboratory protocols in protocols.io to enhance the reproducibility of your results. Protocols.io assigns your protocol its own identifier (DOI) so that it can be cited independently in the future. For instructions see: https://journals.plos.org/plosone/s/submission-guidelines#loc-laboratory-protocols. Additionally, PLOS ONE offers an option for publishing peer-reviewed Lab Protocol articles, which describe protocols hosted on protocols.io. Read more information on sharing protocols at https://plos.org/protocols?utm_medium=editorial-email&utm_source=authorletters&utm_campaign=protocols.

We look forward to receiving your revised manuscript.

Kind regards,

Yung-Fu Chang

Academic Editor

PLOS ONE

3. We note that Figure 1 and 2 in your submission contain copyrighted images. All PLOS content is published under the Creative Commons Attribution License (CC BY 4.0), which means that the manuscript, images, and Supporting Information files will be freely available online, and any third party is permitted to access, download, copy, distribute, and use these materials in any way, even commercially, with proper attribution. For more information, see our copyright guidelines: http://journals.plos.org/plosone/s/licenses-and-copyright.

a. You may seek permission from the original copyright holder of Figure 1 and 2 to publish the content specifically under the CC BY 4.0 license. 

Reviewers' comments:

Reviewer's Responses to Questions

**Comments to the Author**

1. Is the manuscript technically sound, and do the data support the conclusions?

Reviewer #1: Yes

2. Has the statistical analysis been performed appropriately and rigorously? 

Reviewer #1: Yes

3. Have the authors made all data underlying the findings in their manuscript fully available?

Reviewer #1: Yes

4. Is the manuscript presented in an intelligible fashion and written in standard English?

Reviewer #1: Yes

5. Review Comments to the Author

Reviewer #1: The authors describe a method for rapid phage isolation with fewer resource requirements relative to traditional methods. Though less sensitive, the method could be helpful in several circumstances, particularly when time is a concern. In general, the paper is well-written and the methodology used is sound. Though isolating phages in such a manner has been previously reported (using 96-well plates and observing growth curves for inhibition), the authors study adds value by evaluating a wide range of species and assessing several different phage sources. Additionally, they provide a time and cost analysis, which is helpful. I would also note that their results are consistent with many of our own findings in regard to the isolation of phages from wastewater, so I would imagine reproducibility would be high. Overall, the reported method is likely to be frequently adopted by other research groups.

Comments:

L46: Many would argue that this was not the first clinical use of personalized phage therapy, though it is certainly one of the more high-profile cases in the US.

L77: Should report time of year when wastewater was collected. We've found it can have significant impact of the success of phage isolation.

L99: Though it is reported a few paragraphs before, I would suggest saying "filtered wastewater" to make it more obvious to the reader.

L106: Do you mean 0.7% agar?

L134: Sentence is a bit confusing. Meaning assessing the negative samples via conventional plaque assay?

L143: aureus?

L151: The traditional and low-volume growth conditions seem to have only significantly differed by volume (6 mL vs 200 ul enrichments). Any hypotheses as to why this makes a difference for isolate growth? Perhaps gas exchange?

L168: We generally sample the AS directly, and not the RAS. Some additional details of what this type of sample is like may be helpful. Also, were any steps taken to elute phages from the solids? I would expect fairly different microbial (and therefore phage) compositions between DI and AS.

L201: I'm glad the authors mention the use of concentration methods to improve yield, but evaluating this would have been easy and added much value to the study.

Table 2: The number of samples is somewhat confusing between the low-volume and traditional methods. For traditional, 66 isolates x 4 phage sources = 264 combinations. For the low-volume, 8 plates x 96 wells = 768 wells. Were triplicate wells used? If so, 768/3 = 256 samples.

Table 4: For some samples (e.g., Bordetella), phages were found for the low-volume but not traditional methods. Why might that be since the traditional is typically more sensitive (as the authors report)?

6. PLOS authors have the option to publish the peer review history of their article (what does this mean?). If published, this will include your full peer review and any attached files.

Reviewer #1: No

---

## [Author Response · Author response to Decision Letter 0]

28 Jan 2024

Response to Reviewers

Response: Concur with editor comment. We have read guidelines and made appropriate adjustments to the manuscript to adopt PLOSOne’ style requirements. 

Response: Concur with editors. We have created a data set that includes information used to create figures and tables. This dataset has been uploaded at DOI 10.17605/OSF.IO/AHGJF for open access. for open access. This data set contains three main excel files: i) Cost comparison; ii) Growth curves to demonstrate identification of phage from waste water; and iii) High throughput phage discovery. This documentation is in addition to information already available in the manuscript, tables and figures.

3. We note that Figure 1 and 2 in your submission contain copyrighted images. All PLOS content is published under the Creative Commons Attribution License (CC BY 4.0), which means that the manuscript, images, and Supporting Information files will be freely available online, and any third party is permitted to access, download, copy, distribute, and use these materials in any way, even commercially, with proper attribution. For more information, see our copyright guidelines: http://journals.plos.org/plosone/s/licenses-and-copyright.

a. You may seek permission from the original copyright holder of Figure 1 and 2 to publish the content specifically under the CC BY 4.0 license. 

Response: We concur with editors. Figure 1 and Figure 2 were designed and built using an application designated BioRender. Patrick Kenney, co-author of this manuscript, has been granted a license to use the BioRender application, including its content, icons, templates and other original artwork. Accordingly, Figures 1 and 2 now have appropriate release documentation for publication that is added to the resubmission. According to BioRender, this license permits BioRender content to be sublicensed for use in journal publications. All rights and ownership of BioRender content are reserved by BioRender. All completed graphics are accompanied by this citation: “Created with BioRender.com”. BioRender content included in the completed graphic is not licensed for any commercial uses beyond publication in a journal. 

The figure legends in the initial submission were removed from all of the *.tiff files to ensure there was only one copy of the legend and avoid that way accidental inconsistency. Figure legends are included at the end of the main manuscript. 

Response: We have reviewed the references list and we confirm that all information is accurate. 

Reviewers' comments:

Reviewer #1: The authors describe a method for rapid phage isolation with fewer resource requirements relative to traditional methods. Though less sensitive, the method could be helpful in several circumstances, particularly when time is a concern. In general, the paper is well-written and the methodology used is sound. Though isolating phages in such a manner has been previously reported (using 96-well plates and observing growth curves for inhibition), the authors study adds value by evaluating a wide range of species and assessing several different phage sources. Additionally, they provide a time and cost analysis, which is helpful. I would also note that their results are consistent with many of our own findings in regard to the isolation of phages from wastewater, so I would imagine reproducibility would be high. Overall, the reported method is likely to be frequently adopted by other research groups.

Response: The authors thank this reviewer for the thorough and thoughtful comments – they are much appreciated!

L46: Many would argue that this was not the first clinical use of personalized phage therapy, though it is certainly one of the more high-profile cases in the US.

Response: Concur with reviewer comment. Accordingly, the sentence in question: “The first clinical use of personalized phage therapy was in the treatment of a multidrug-resistant Acinetobacter baumannii infection in 2016” was changed to read: “A highly popularized clinical use of personalized phage therapy was in the treatment of a multidrug-resistant Acinetobacter baumannii infection in 2016”

L77: Should report time of year when wastewater was collected. We've found it can have significant impact of the success of phage isolation.

Response: Appreciate reviewer comment. Accordingly, Line 77 was adjusted to read, “In April 2023, a local wastewater treatment facility…”

L99: Though it is reported a few paragraphs before, I would suggest saying "filtered wastewater" to make it more obvious to the reader.

Response. Agree with reviewer. Accordingly, we have edited the phrase to read: “filtered wastewater”

L106: Do you mean 0.7% agar?

Response: Appreciate reviewer recognition of typo. The answer is Yes. The phrase was edited and it now reads: “0.7% agar”, not 70%.

L134: Sentence is a bit confusing. Meaning assessing the negative samples via conventional plaque assay?

Response: Appreciate reviewer comment. Accordingly, we edited the sentence. The sentence in question: “At a species level, only coagulase-negative Staphylococcus, Vibrio cholerae, and Yersinia enterocolitica did not have lytic phage detected during testing,” was change to: “Coagulase-negative Staphylococcus, Vibrio cholerae, and Yersinia enterocolitica were the only species without plaques found in conventional plaque assays.”

L143: aureus?

Response: This isolate was a non-Staphylococcus aureus. Conventional microbiologic testing did not differentiate the species further, so it is in our biobank simply as “Coagulase-negative Staphylococcus”

L151: The traditional and low-volume growth conditions seem to have only significantly differed by volume (6 mL vs 200 ul enrichments). Any hypotheses as to why this makes a difference for isolate growth? Perhaps gas exchange?

Response: Appreciate reviewer question. Although, we have no experimental evidence regarding oxygen levels of culture environments with the traditional vs low-volume methods, we agree with reviewer that differential gas-exchange may have affected bacterial growth. In fact, the isolates that had poor growth were more anaerobic in nature. Therefore, the impairment of gas exchange at the lower parts of the test tubes, compared to a shallow wells of 96-well plates, likely promoted growth of those species. The differences in the low volume samples between DI and RAS are less clear and likely related to some undefined inhibitory components in the RAS. A note to this regard was placed in this section of the text.

L168: We generally sample the AS directly, and not the RAS. Some additional details of what this type of sample is like may be helpful. Also, were any steps taken to elute phages from the solids? I would expect fairly different microbial (and therefore phage) compositions between DI and AS.

Response: Appreciate reviewer comment. Obtaining RAS vs AS was facilitated by the water plant facility and the processing of both samples was out of interest scientific curiosity. We did intentionally obtain both RAS and DI to allow for greater microbial diversity. In particular, aerobes are only rarely isolated in RAS compared to DI. We did not use elution steps, but we did ensure that solids were present in solution right up until the point of centrifugation/filtration for a day’s experiments to hopefully allow for better capture. A sentence regarding differences between RAS and DI was added to the text.

L201: I'm glad the authors mention the use of concentration methods to improve yield, but evaluating this would have been easy and added much value to the study.

Response: We thank reviewer for the comment. Concentration methods would have been ideal, and it was in fact attempted. Unfortunately, time would not allow for concentration steps. We did not have an equipment setup for TFF or ultracentrifugation, and our suppliers were quoting multi-month delays for ultrafiltration tubes in the post-COVID supplies crunch. 

Table 2: The number of samples is somewhat confusing between the low-volume and traditional methods. For traditional, 66 isolates x 4 phage sources = 264 combinations. For the low-volume, 8 plates x 96 wells = 768 wells. Were triplicate wells used? If so, 768/3 = 256 samples.

Response: 8 plates is an artifact of the amount of effort involved in simultaneous setup; I found that trying to manage more than 16 bacterial isolates in the traditional method was too labor intensive, so I limited each plate to 16 bacterial isolates and 2 waste conditions.

Table 4: For some samples (e.g., Bordetella), phages were found for the low-volume but not traditional methods. Why might that be since the traditional is typically more sensitive (as the authors report)?

Response: Appreciate reviewer question. We do not have a good explanation for this finding. One possible explanation with no experimental evidence is that low-volume conditions increase environmental stress of Bartonella which may result in upregulation of genes mediating lytic cycle of Bordetella-associated phages. Traditional method, using larger liquid media volume and less environmental stress, may downregulate genes mediating lytic cycle of Bordetella-associated phages rendering no free phages to be detected in media.

---

## [Decision Letter · Decision Letter 1]

31 Jan 2024

Low-Volume Enrichment Method Supports High Throughput Bacteriophage Screening and Isolation from Wastewater

PONE-D-23-39330R1

Dear Dr. Gomez-Duarte,

We’re pleased to inform you that your manuscript has been judged scientifically suitable for publication and will be formally accepted for publication once it meets all outstanding technical requirements.

Kind regards,

Yung-Fu Chang

Academic Editor

PLOS ONE

Additional Editor Comments (optional):

Reviewers' comments:

Reviewer's Responses to Questions

**Comments to the Author**

1. If the authors have adequately addressed your comments raised in a previous round of review and you feel that this manuscript is now acceptable for publication, you may indicate that here to bypass the “Comments to the Author” section, enter your conflict of interest statement in the “Confidential to Editor” section, and submit your "Accept" recommendation.

Reviewer #1: All comments have been addressed

2. Is the manuscript technically sound, and do the data support the conclusions?

Reviewer #1: Yes

3. Has the statistical analysis been performed appropriately and rigorously? 

Reviewer #1: Yes

4. Have the authors made all data underlying the findings in their manuscript fully available?

Reviewer #1: Yes

5. Is the manuscript presented in an intelligible fashion and written in standard English?

Reviewer #1: Yes

6. Review Comments to the Author

Reviewer #1: Thanks for supplying answers to all of my questions. The shortage of TFF cassettes was felt by us as well, so completely understandable. Great work!

7. PLOS authors have the option to publish the peer review history of their article (what does this mean?). If published, this will include your full peer review and any attached files.

Reviewer #1: No

---

## [Editor Report · Acceptance letter]

19 Feb 2024

PONE-D-23-39330R1 

PLOS ONE

Dear Dr. Gomez-Duarte, 

I'm pleased to inform you that your manuscript has been deemed suitable for publication in PLOS ONE. Congratulations! Your manuscript is now being handed over to our production team.

Kind regards, 

on behalf of

Dr. Yung-Fu Chang 

Academic Editor

PLOS ONE